The effects of phylogeny, body size, and locomotor behavior on the three-dimensional shape of the pelvis in extant carnivorans

Lewton Kristi L. kristilewton@gmail.com 1 2 3
Brankovic Ryan 2
Byrd William A. 1 4
Cruz Daniela 2
Morales Jocelyn 2
Shin Serin 5
1 Department of Integrative Anatomical Sciences, University of Southern California , Los Angeles , CA , United States of America
2 Department of Biological Sciences, University of Southern California , Los Angeles , CA , United States of America
3 Department of Mammalogy, Natural History Museum of Los Angeles , Los Angeles , CA , United States of America
4 Department of Life Sciences, Santa Monica College , Santa Monica , CA , United States of America
5 North Hollywood High School , North Hollywood , CA , United States of America
Farke Andrew
Electronic publication date: 2020 Feb 20
Publication date: 2020
Volume: 8
Electronic Location ID: e8574
Received 2019 Nov 11; Accepted 2020 Jan 15
Copyright: ©2020 Lewton et al.
Copyright year: 2020
Copyright holder: Lewton et al.
License: This is an open access article distributed under the terms of the Creative Commons Attribution License, which permits unrestricted use, distribution, reproduction and adaptation in any medium and for any purpose provided that it is properly attributed. For attribution, the original author(s), title, publication source (PeerJ) and either DOI or URL of the article must be cited.
License URL: https://creativecommons.org/licenses/by/4.0/

Keywords: 3D geometric morphometrics, Pelvis, Functional morphology, Anatomy, Scaling, Phylogenetic comparative methods

Funding: University of Southern California This work was supported by the University of Southern California. The funders had no role in study design, data collection and analysis, decision to publish, or preparation of the manuscript.

==============================
The mammalian pelvis is thought to exhibit adaptations to the functional demands of locomotor behaviors. Previous work in primates has identified form-function relationships between pelvic shape and locomotor behavior; few studies have documented such relationships in carnivorans, instead focusing on long bones. Most work on the functional morphology of the carnivoran pelvis, in particular, has used univariate measures, with only a few previous studies incorporating a three-dimensional (3D) analysis. Here we test the hypothesis that carnivoran taxa that are characterized by different locomotor modes also differ in 3D shape of the os coxae. Using 3D geometric morphometrics and phylogenetic comparative methods, we evaluate the phylogenetic, functional, and size-related effects on 3D pelvis shape in a sample of 33 species of carnivorans. Using surface models derived from laser scans, we collected a suite of landmarks (N = 24) and curve semilandmarks (N = 147). Principal component analysis on Procrustes coordinates demonstrates patterns of shape change in the ischiopubis and ilium likely related to allometry. Phylogenetic generalized least squares analysis on principal component scores demonstrates that phylogeny and body size have greater effects on pelvic shape than locomotor function. Our results corroborate recent research finding little evidence of locomotor specialization in the pelvis of carnivorans. More research on pelvic morphological integration and evolvability is necessary to understand the factors driving pelvic evolution in carnivorans.

Introduction

Identifying associations between skeletal form and locomotor function are critical for determining how skeletons adapt to the biological roles that they must perform. These form-function links are especially important for reconstructing locomotor behaviors in extinct species (Rudwick, 1964; Ross et al., 2002). The ossa coxae are a crucial component of the locomotor system because they provide anchorage for the muscles that propel the body during locomotion and they transmit forces from the hindlimb to the torso (Dalstra & Huiskes, 1995). However, the precise relationships between mammalian pelvic form and locomotor function are not well understood because previous studies of mammalian functional morphology have focused primarily on the long bone elements of the fore- and hindlimbs (e.g., Van Valkenburgh, 1987; Schutz & Guralnick, 2007; Lewis & Lague, 2010; Polly, 2010; Fabre et al., 2013; Samuels, Meachen & Sakai, 2013). Much of the previous work on the mammalian pelvic skeleton in particular has centered on gene expression underlying the embryological development of the ilium, ischium, and pubis (Pellegrini et al., 2001; Pomikal & Streicher, 2010), and general associations between linear measures of pelvic elements and locomotor behavior or ecomorphology (Davis, 1964; Barry, 1976; Taylor, 1976). Pelvic skeletal functional morphology is more commonly investigated in human and nonhuman primates as a foundation for reconstructing the evolution of bipedality in the hominin lineage and the locomotor behaviors of fossil apes and monkeys (e.g., Berge, 1984; Ward, 1993; Lewton, 2015a; Lewton, 2015b; Hammond & Almécija, 2017; Ward, Maddux & Middleton, 2018).

This previous research in primates has demonstrated several key features of the primate pelvis that are adaptations to locomotion (i.e., that differ according to the biomechanical requirements of different locomotor modes), including the dimensions of the iliac blade (e.g., width, Ward, 1991; Ward, Maddux & Middleton, 2018), lower ilium (height and cross-sectional area, e.g., Lewton, 2015a; Lewton, 2015b; Hammond & Almécija, 2017), ischium (e.g., ischial length, Fleagle & Anapol, 1992; Lewton & Scott, 2017), and pubis (Lewton, 2015a; Lewton, 2015b; Lewton & Dingwall, 2016). This previous research has incorporated both univariate and three-dimensional geometric morphometric data and found the length of the lower ilium to be particularly informative, indicating that larger-bodied primates and/or taxa that encounter relatively large locomotor loads (i.e., large external forces—such as ground reaction forces—that result from locomotor behavior) tend to have shorter, more robust lower ilia presumably to maintain rigidity with increasing mechanical stresses (Lewton, 2015a; Lewton, 2015b). Relatively short and broad ilia are associated with species that use more orthograde postures and that encounter relatively large locomotor loads, such as bipedal hominins (Le Gros Clark, 1955; Robinson, 1972; Badoux, 1974; Leutenegger, 1974; Lovejoy et al., 2009; Lewton, 2015a; Lewton, 2015b) and large-bodied vertical clinging and leaping strepsirrhines (Lewton, 2015a; Lewton, 2015b). Primate ischiopubic morphology is similarly reflective of mechanical needs; species that encounter relatively large locomotor loads have long pubic symphyses and short pubic rami (Howell, 1944; Ward, 1991; Anemone, 1993; Lewton, 2015a; Lewton, 2015b).

Compared to research on primate pelvic skeletal morphology, less work has been conducted on the functional morphology of the carnivoran pelvis. Previous research on the functional aspects of the carnivoran pelvis has focused primarily on univariate and two-dimensional analyses. Early work on carnivoran pelvic morphology in relationship to locomotor function yielded mixed results; differences among locomotor groups in pelvic morphology were identified (Davis, 1964; Barry, 1976; Taylor, 1976), but the functional relevance of these differences was not well understood (Davis, 1964). Combining kinematic (from cineradiographs) and limited morphological data (acetabular coverage of the femoral head) from procyonids, felids, and canids, Jenkins & Camazine (1977) found functional relationships between the position and angular excursion of the femur during locomotion with articular morphologies of the femoral head; for example, cursorial carnivorans exhibit morphologies that restrict hip abduction capabilities to maintain limb movements in a parasagittal plane. More recently, Martín-Serra, Figueirido & Palmqvist (2014a) used 3D geometric morphometric methods to investigate the effects of locomotion on pelvic morphology. Using 16 landmarks on the ilium, ischium, and pubis, Martín-Serra, Figueirido & Palmqvist (2014a) found significant effects of phylogeny and body size on pelvic morphology, but the effect of locomotor behavior was less clear as locomotor behaviors were correlated with phylogeny. However, Martín-Serra, Figueirido & Palmqvist (2014a) captured some, but not all, aspects of pelvic shape, as their study did not use semilandmarks and, as a result, did not record the shape of the prominent curves of the pelvis such as the iliac crest, the arcuate line, or the shape of the ischiopubis, and they did not include representatives of herpestids, mephitids, or viverrids. These data would be informative because they provide information related to the shape of prominent regions of attachment for muscles that are involved in propulsion of the hindlimb and in flexion and extension of the spine (e.g., the hindlimb extensors along the ilium and ischium and the erector spinae muscles along the medial aspect of the iliac crest, respectively). These bony regions have been shown in other mammals to correlate with locomotor behavior and adaptation (e.g., Lewton, 2015b; Ward, Maddux & Middleton, 2018). Furthermore, the inclusion of species of herpestids, mephitids, and viverrids is important because it allows an investigation of the effects of body size on pelvic shape by including more carnivoran taxa that are small-bodied, and it also expands the locomotor and postural behaviors in the sample (e.g., including semifossorial species). Therefore, the aim of this paper is to test functional hypotheses of pelvic form in the Order Carnivora using a phylogenetically-diverse sample and 3D shape data.

Methods

The osteological sample comprises ossa coxae of 56 specimens of 33 species from 10 families of Carnivora from the Natural History Museum of Los Angeles (Table 1). To increase sample sizes per taxon, we included both captive and wild specimens and used a mixed-sex sample (see Supplemental Information). The locomotor behavior of each taxon was categorized into one of six locomotor groups (arboreal, cursorial, natatorial, scansorial, semifossorial, or terrestrial). Locomotor behavior classifications were derived from the literature (see references in Table 1). Data were collected on the right os coxae of adult specimens (judged by pelvic epiphyseal fusion).

Table 1 Sample size and locomotor category for each taxon (N = 33).

Taxon	N	Locomotion	Behavioral reference	
Ailuridae				
Ailurus fulgens	2	Arboreal	Nowak (2005) and Roberts & Gittleman (1984)	
Canidae				
Canis latrans	2	Cursorial	Bekoff (1977)	
Nyctereutes procyonoides	1	Terrestrial	Ward & Wurster-Hill (1990)	
Otocyon megalotis	2	Cursorial	Morlo, Gunnell & Nagel (2010)	
Urocyon cinereoargenteus	2	Scansorial	Trapp & Hallberg (1975)	
Urocyon littoralis	5	Terrestrial	Moore & Collins (1995)	
Vulpes vulpes	2	Cursorial	Nowak (2005)	
Felidae				
Acinonyx jubatus	1	Cursorial	Sunquist & Sunquist (2002) and Wilson et al. (2013)	
Felis silvestris	1	Scansorial	Meachen-Samuels & Van Valkenburgh (2009) and Sunquist & Sunquist (2002)	
Leptailurus serval	1	Terrestrial	Meachen-Samuels & Van Valkenburgh (2009) and Sunquist & Sunquist (2002)	
Lynx canadensis	2	Scansorial	Van Valkenburgh (1987)	
Lynx rufus	3	Scansorial	Meachen-Samuels & Van Valkenburgh (2009) and Sunquist & Sunquist (2002)	
Otocolobus manul	1	Terrestrial	Meachen-Samuels & Van Valkenburgh (2009) and Sunquist & Sunquist (2002)	
Panthera leo	1	Terrestrial	Meachen-Samuels & Van Valkenburgh (2009) and Sunquist & Sunquist (2002)	
Panthera pardus	1	Scansorial	Sunquist & Sunquist (2002) and Van Valkenburgh (1987)	
Panthera tigris	2	Terrestrial	Meachen-Samuels & Van Valkenburgh (2009) and Sunquist & Sunquist (2002)	
Prionailurus bengalensis	1	Scansorial	Meachen-Samuels & Van Valkenburgh (2009) and Sunquist & Sunquist (2002)	
Puma concolor	2	Scansorial	Sunquist & Sunquist (2002) and Van Valkenburgh (1987)	
Herpestidae				
Atilax paludinosus	2	Natatorial	Baker (1992) and Nowak (2005)	
Cynictis penicillata	2	Semifossorial	Taylor & Meester (1993)	
Galerella pulverulenta	1	Terrestrial	Nowak (2005)	
Galerella sanguinea	1	Terrestrial	Nowak (2005) and Taylor (1976)	
Herpestes ichneumon	1	Terrestrial	Nowak (2005) and Taylor (1976)	
Mungos mungo	1	Terrestrial	Nowak (2005) and Taylor (1976)	
Hyaenidae				
Proteles cristata	3	Terrestrial	Koehler & Richardson (1990)	
Mephitidae				
Mephitis mephitis	2	Semifossorial	Wade-Smith & Verts (1982)	
Mustelidae				
Eira barbara	1	Scansorial	Presley (2000)	
Nandiniidae				
Nandinia binotata	1	Arboreal	Nowak (2005) and Taylor (1976)	
Procyonidae				
Procyon lotor	3	Scansorial	Nowak (2005)	
Ursidae				
Melursus ursinus	1	Scansorial	Nowak (2005) and Van Valkenburgh (1987)	
Ursus americanus	2	Scansorial	Nowak (2005)	
Viverridae				
Arctictis binturong	2	Arboreal	Nowak (2005)	
Paradoxurus sp.	1	Arboreal	Nowak (2005)	

Three-dimensional os coxae models were constructed from laser scans using a NextEngine HD Laser Scanner (NextEngine, Inc., Santa Monica). Scan settings varied depending on the size of the specimen, where small specimens were scanned in macro mode, and larger specimens in wide mode. High definition (HD) settings and 12-13 rotations were used for all scans. Specimens were scanned in two to three orientations to ensure adequate capture of the entire surface. Scans were exported as polygon (.ply) files and were digitally aligned and merged in Geomagic Wrap software (3D Systems, Inc., Morrisville, NC). Surface models were then processed in Geomagic, which included removing spikes and filling small holes in the mesh. The resulting .ply files were imported into Checkpoint software (Stratovan Corp., Davis) and a suite of 3D landmarks (N = 24) and curve semilandmarks (N = 147) were digitally extracted from each model (Fig. 1, Table 2; the raw, unadjusted landmarks for all specimens are provided in the Supplemental Information). Landmarks reflect homologous anatomical locations based on muscle attachments, joint articulations, loci of epiphyseal fusion, or other regions of anatomical interest following Lewton (2012) and Lewton (2015b). Semilandmarks were placed along the following eight curves: (1) iliac crest, (2) arcuate line, (3) dorsal iliac margin, (4) acetabular lunate surface rim—external margin, (5) acetabular lunate surface rim—internal margin, (6) ischiopubic ramus margin, (7) lateral iliac margin, (8) obturator foramen margin (Fig. 1). One specimen (LACM 90728) had a small hole in the acetabular notch that precluded placement of Landmark 11, so this landmark was estimated using the estimate.missing function in the ‘geomorph’ package (Adams, Collyer & Kaliontzopoulou, 2019) for R software (R Core Team, 2019).

Figure 1 Three-dimensional landmarks are shown in ventral (A), lateral (B), and dorsal (C) views on a fox (Vulpes Vulpes) os coxae.

Yellow labeled points indicate landmarks, while blue points and black lines indicate semilandmark curves. Landmark and curve definitions are listed in Table 2.

Table 2 Three-dimensional landmark and curve definitions.

No.	Name	Definition	Type	
L1	ASIS	The anterior-most point on the lateral extent of the iliac crest (anterior superior iliac spine); site of attachment for m. sartorius (Lewton, 2012; Lewton, 2015a)	II	
L2	AIIS	The anterior-most point on the anterior inferior iliac spine. If only a bony roughening, the point in the center of the AIIS rugosity; site of attachment for m. rectus femoris (Lewton, 2012; Lewton, 2015a)	II	
L3	Lateral ilium	The lateral-most point on the lateral aspect of the iliac margin, above the AIIS, where the cross-section of the lower ilium is smallest (Lewton, 2012; Lewton, 2015a)	III	
L4	PSIS	The superomedial-most point on the posterior iliac crest (Lewton, 2012; Lewton, 2015a)	II	
L5	Inferior auricular surface	The inferior-most extent of the auricular surface, on the dorsal aspect of the pelvis (Lewton, 2012; Lewton, 2015a)	II	
L6	Dorsal ilium	The dorsal-most point on the dorsal aspect of the lower ilium, where the cross-section of the lower ilium is smallest. Taken directly across from Landmark 3 (Lewton, 2012; Lewton, 2015a)	III	
L7	Ischial spine	The dorsal-most projection of the spine located on the posterior ischium, medial to the acetabulum (Lewton, 2012; Lewton, 2015a)	II	
L8	Ischial tuberosity	The dorsal-most point on the posterior ischium, medial to the acetabulum (Lewton, 2012; Lewton, 2015a)	II	
L9	Superior acetabulum	The point on the superior rim of the acetabulum that marks the intersection of the iliac margin and acetabulum, which is defined as the extension of the line connecting ASIS and AIIS (Lewton, 2012; Lewton, 2015a)	III	
L10	Inferior acetabulum	The point on the inferior rim of the acetabulum directly across from Landmark 9, along the long axis of the ischium (Lewton, 2012; Lewton, 2015a)	III	
L11	Mid-acetabulum	The center of the acetabulum; defined as the midpoint of the line between Landmarks 9 and 10 (Lewton, 2012; Lewton, 2015a)	III	
L12	Ischium	The distal-most point on the ischium that forms a line with the center of the acetabulum that is parallel to the long axis of the ischium (Lewton, 2012; Lewton, 2015a)	III	
L13	Superior pubic symphysis	The superior-most point on the pubic symphysis, taken on the most medial point of the pubis (Lewton, 2012; Lewton, 2015a)	II	
L14	Inferior pubic symphysis	The inferior-most point on the pubic symphysis, taken on the most medial point of the pubis (Lewton, 2012; Lewton, 2015a)	II	
L15	Lateral sacrum	The point that marks the intersection of the arcuate line of the ilium and the sacrum (Lewton, 2012; Lewton, 2015a)	I	
L16	Transverse diameter of pelvis	The point on the arcuate line that constitutes the maximum distance between the arcuate line of the opposing os coxa (Tague, 2005; Lewton, 2012; Lewton, 2015a)	II	
L17	Medial ilium	The medial-most point on the medial aspect of the lower ilium, where the cross-section of the ilium is the smallest. Taken directly across from Landmarks 3 and 6 (Tague, 2005; Lewton, 2012; Lewton, 2015a)	III	
L18	Pectineal tuberosity	Maximum projection of the pectineal tuberosity (Álvarez, Ercoli & Prevosti, 2013)	I	
L19	Obturator foramen 1	Cranial end of the major axis of the obturator foramen	II	
L20	Obturator foramen 2	Caudal end of the major axis of the obturator foramen	II	
L21	Obturator foramen 3	Cranial end of the minor axis of the obturator foramen	II	
L22	Obturator foramen 4	Caudal end of the minor axis of the obturator foramen	II	
L23	Cranial lunate	Ventral-most point on the cranial lunate horn	II	
L24	Caudal lunate	Ventral-most point on the caudal lunate horn	II	
C1	Iliac crest	Curve from points 1 to 4	Semilandmark curve	
C2	Arcuate line	Curve from points 15 to 13	Semilandmark curve	
C3	Dorsal ilioischial curve	Curve from the piriformis tubercle cranially to point 8	Semilandmark curve	
C4	Acetabular rim—external	Curve from points 23 to 24, on the external aspect of the acetabular rim	Semilandmark curve	
C5	Acetabular rim—internal	Curve from points 23 to 24, on the internal aspect of the acetabular rim	Semilandmark curve	
C6	Ischiopubic ramus	Curve from points 8 to 14	Semilandmark curve	
C7	Lateral iliac margin	Curve from points 1 to 9	Semilandmark curve	
C8	Obturator foramen	Curve from points 19 to 22	Semilandmark curve	
Notes.

L landmark

C curve

All landmarks were placed on surface models by a single observer (WAB). Intraobserver landmark error was assessed by repeating the landmarking process five times on a single specimen and calculating the percent error for each landmark. The average error over all landmarks was 1.98%. Only one landmark had an error rate over 5% (Landmark 11, the center of the acetabulum, 7% error).

Geometric morphometric methods were used to test hypotheses of shape differences among locomotor groups. Landmark configurations were scaled, rotated, and translated using Generalized Procrustes Analysis. The criterion used for sliding semilandmarks along curves was minimizing bending energy (Gunz & Mitteroecker, 2013). Species means of Procrustes coordinates were computed and principal component analysis (PCA) was performed using a phylomorphospace approach.

All phylogenetic comparative analyses used a tree derived from Nyakatura and Bininda-Emonds’s (2012) Carnivora supertree based on molecular data (available in the Supplemental Information). The treedata function in the ‘geiger’ package (Harmon et al., 2008) for R software (R Core Team, 2019) was used to ensure that the species mean principal component (PC) scores and the tree topology were concordant. The effects of locomotor behavior and body size on pelvic shape were tested using phylogenetic generalized least squares analysis (PGLS). Because pelvis size correlates with body size (e.g., Ward, 1991; Lewton, 2010; Lewton, 2015a; Ward, Maddux & Middleton, 2018), centroid size of the landmark configuration was used as an estimate of overall body size. The PGLS regressions take the form of PC score ∼ locomotion + centroid size, where “locomotion” is a categorical variable with six levels. Degree of phylogenetic signal in the Procrustes coordinates was assessed using the multivariate K-statistic (Adams, 2014). All geometric morphometric analyses were conducted using the ‘geomorph’ (Adams, Collyer & Kaliontzopoulou, 2019) package for R software (see Supplemental Information for R code and files).

Results

Principal component analysis and phylomorphospace

The first four principal components (PCs) describe 74% of the variation in this sample. PC 1 reflects variation in ischiopubic shape and describes 31% of sample variation. PC 2 relates to ilium width and orientation and describes 23% of sample variation. Figure 2A plots PC 2 on PC 1 and shows some separation in phylomorphospace among taxonomic groups. Along PC 1, species with more negative PC 1 scores demonstrate triangular pubic bones, with longer pubic rami and shorter pubic symphyses; taxa that exemplify more triangular pubic bones are skunks (Fig. 2B). Other taxa with negative PC 1 scores include ursids, civets, raccoons, aardwolves, and some herpestids. Species with more positive PC scores demonstrate square-shaped pubic bones, with shorter iliopubic and ischiopubic rami and longer pubic symphyses; taxa that exemplify more square-shaped pubic bones are felids and canids (Fig. 2B) . Along PC 2, species with more negative PC scores (e.g., ursids, aardwolves, raccoons, and coyotes) exhibit wider and more laterally-flaring iliac blades, while species with more positive PC 2 scores (e.g., civets and most herpestids) exhibit narrow ilia with the iliac blade oriented in a parasagittal plane (Fig. 2B).

Figure 2 Phylomorphospace plots and representative os coxae shapes.

Phylomorphospace plots for PC1 on PC2 (A) and PC3 on PC4 (B). The phylogeny is shown in grey, points correspond to species means PC scores, colored by locomotor category. Legend in (C) follows (A). The os coxae shapes represented by the extremes of the PC axes are shown in (B) and (D), corresponding to plots (A) and (C), respectively.

Variation in phylomorphospace among taxa in the PC 3 vs PC 4 plot generally corresponds to family (Fig. 2C); shape variation along these axes is minimal (Fig. 2D). PC 3 reflects variation in overall length of the ilium and ischium and describes 13.5% of sample variation. Canids generally have positive PC3 scores (reflecting shorter ischia and longer ilia), while felids and herpestids generally have negative PC3 scores (reflecting longer ischia and shorter ilia, (Figs. 2C, 2D). PC 4 reflects variation in ilium width and orientation and ischium breadth and describes 6.5% of sample variation. Palm civets are separated from all other taxa at the positive end of PC 4, reflecting wide ilia that are slightly more parasagittally-aligned, and broader ischia (Figs. 2C, 2D). Other taxa with positive PC 4 scores include most of the felids and canids (but notably not the most terrestrial felids, the manul, serval, and cheetah). Taxa with negative PC 4 scores (reflecting narrower ilia and more slender ischia) include herpestids, skunks, and the most terrestrial felids.

Phylogenetic generalized least squares analysis

The degree of phylogenetic signal in the species-mean Procrustes shape variables is statistically significant (K = 0.52, p = 0.001). The PGLS analysis demonstrated that neither locomotor mode nor size had a significant effect on PC 1 (F = 0.72, p = 0.60 and F = 1.41, p = 0.25, respectively) and that locomotor mode did not have a significant effect on PC 2 and PC 3 (F = 2.37, p = 0.07 and F = 1.32, p = 0.28, respectively). Size, however, did have a significant effect on PC 2 and PC 3 (F = 30.3, p = 0.0001 and F = 6.53, p = 0.02, respectively). For PC 4, the PGLS model demonstrated that neither locomotor mode nor size were significant (F = 2.35, p = 0.08 and F = 1.36, p = 0.26, respectively). The correlation between PC 2 and log-centroid size is negative (r =  − 0.67), indicating that taxa located at the negative end of the PC 2 axis are larger than those at the positive end (Fig. 2A). Although the correlation between PC 3 and log-centroid size is statistically significant (r = 0.41), a bivariate plot indicates no relationship between these two variables.

Discussion

We tested the hypothesis that carnivorans that differ in habitual locomotor behavior would exhibit significantly different 3D pelvic shapes. This hypothesis was not supported. Although carnivorans demonstrate variation in pelvic shapes, the primary factors influencing shape are phylogeny and size, not locomotor function. These results are somewhat surprising given documented differences in pelvic morphology due to locomotion and positional behavior in other mammals (Ward, 1991; Anemone, 1993; Álvarez, Ercoli & Prevosti, 2013; Lewton, 2015a; Lewton, 2015b; Lewton & Dingwall, 2016; Tague, 2019).

Previous research on carnivoran pelvic shape has primarily focused on two-dimensional (2D) geometric morphometrics (Álvarez, Ercoli & Prevosti, 2013) and univariate analyses of linear measures (Davis, 1964; Barry, 1976; Taylor, 1976). Using 2D pelvic landmarks on a sample of nine orders of mammals, Álvarez, Ercoli & Prevosti (2013) documented variation in acetabulum size, ventral elongation of the pubis, the length of the pubic symphysis, and dorsal projection of the ischial tuberosity. Variation in ilium morphology was not fully captured by this previous work because only two landmarks were placed on the ilium (Álvarez, Ercoli & Prevosti, 2013). As in our study, significant phylogenetic signal was found in 2D pelvis shape (Álvarez, Ercoli & Prevosti, 2013), although these authors also suggested that variation in pelvic shape was related to speed of locomotion, with high-speed taxa demonstrating a long ischium and pubic symphysis. Similarly, Barry’s (1976) analysis of univariate pelvic dimensions demonstrated speed-related morphological variation, with separation between cursorial and ambulatory taxa.

In a recent study using 3D geometric morphometric methods on pelvic bones, Martín-Serra, Figueirido & Palmqvist (2014a) found that phylogeny and body size have greater effects on pelvic shape than locomotor behavior. Our study expanded upon Martín-Serra and colleagues’ by sampling from additional carnivoran families, incorporating more landmarks, and including semilandmarks to capture pelvic curves. It is important to note, however, that Martín-Serra, Figueirido & Palmqvist’s (2014a) sample was larger and included more taxonomic and behavioral diversity within some families (canids, hyaenids, mustelids, and ursids) than the present sample. Our results corroborate those of Martín-Serra and colleagues: within carnivorans, pelvis anatomy is more closely related to phylogeny and body size than to locomotor behavior. Parallel analyses on the carnivoran forelimb by Martín-Serra, Figueirido & Palmqvist (2014b) also find that morphological differences are driven by phylogenetic constraint and body size and not locomotor behavior.

Nevertheless, shape variation in our principal component analyses shows some of the same morphological patterns, particularly in the pubis and ilium, as previous work in primates (Lewton, 2015b). In strepsirrhines, scores along the first principal component also describe a spectrum in pubic bone morphology from square to triangular, in which square pubic bones have relatively short pubic rami and long pubic symphyses, and triangular pubic bones have relatively long pubic rami and short symphyses (Lewton, 2015b). In primates, these differences in the shape of the pubic bones were related to differences in body size and potentially locomotor load magnitudes, in which small-bodied primates exhibited elongation of the pubic bones and large-bodied primates exhibited short and broad pubic bones. Variation along the second principal component in primates related to differences in arboreal versus leaping behaviors, in which arboreal primates exhibited small acetabulae and long ischia and pubic symphyses, while leaping primates exhibited the opposite pattern (Lewton, 2015b).

Similarly, the variation in ilium width demonstrated here echoes patterns exhibited in primates. Across primates, ilium width scales with positive allometry (Lewton, 2015a; Middleton et al., 2017; Ward, Maddux & Middleton, 2018), and large-bodied orthograde species (such as indriids and hominoids) exhibit wider ilia than expected (Lewton, 2015a; Lewton, 2015b). A comparison of locomotor groups within strepsirrhines shows that large-bodied vertical clingers and leapers have relatively wider ilia than their small-bodied counterparts and arboreal quadrupedal sister taxa (Lewton, 2015b). In the present study, the widest ilia belong to ursids, a finding noted by other researchers (e.g., Davis, 1964; Martín-Serra et al., 2015) and attributed to “peculiarities, still unknown, in the abdominal wall muscles” (Davis, 1964, p. 110). Given that ursids are some of the largest taxa in this sample, their wide ilia lend further support to the previous findings of positive allometry in ilium width in primates. Although ursids do not habitually use orthograde postures, they are capable of assuming orthograde postures (Davis, 1964) and their pelvic shape may be indicative of the ability to accommodate facultative orthogrady (see also Russo & Williams, 2015). Nevertheless, the functional relevance of wide ilia to orthogrady is not widely supported within our sample of carnivorans.

Our finding of a lack of locomotor effect on pelvic morphology could result from possible limitations of our study. We aimed to increase taxonomic sampling and thus include a mixed-sex sample of a large number of taxa, but relatively small samples per taxon. In addition, we used both captive- and wild-reared animals in an effort to increase sample sizes. Although some differences in long bone articular surface areas have been found in the proximal tibial and distal ulnar joint surfaces between captive- versus wild-reared chimpanzee individuals (Lewton, 2017), consistent differences in postcranial morphology between captive and wild specimens are not documented (Bello-Hellegouarch et al., 2013; Turner et al., 2016). Previous work has not identified broad patterns of sexual dimorphism in pelvic shape or size across carnivorans, although Schutz et al. (2009) found that pelvic shape and size is dimorphic in the small-bodied island fox, Urocyon littoralis. Although an examination of intraspecific pelvic variation is beyond the scope of this study, future work could investigate the effects of sex and rearing on the shape of the carnivoran pelvis.

Given recent work demonstrating a lack of locomotor adaptive signal in the 3D morphology of the carnivoran pelvic girdle, future research that delves into patterns of limb girdle modularity, evolvability, and constraint is needed to determine whether evolutionary constraint limits pelvis evolutionary flexibility (e.g., Marroig et al., 2009) to adapt to different locomotor regimes. It has been suggested that the carnivoran pelvis is modular, with the ilium, ischium, pubis, and acetabulum representing four modules, and that carnivoran families differ in their patterns of pelvic modularity (Martín-Serra, Figueirido & Palmqvist, 2018). However, patterns of morphological integration of pelvic features and estimates of evolvability of pelvic shape have not been investigated in carnivorans. Recent work on patterns of morphological integration between and within limbs in carnivoran demonstrates that cursors have higher levels of morphological integration than non-cursors (Martín-Serra et al., 2015), suggesting that having other functional roles of the forelimb beyond terrestrial locomotion has resulted in increased responsiveness to functional selection pressures in non-cursor carnivorans. Therefore, lack of pelvic variation in carnivoran cursors may be related to increased levels of limb integration. Alternatively, one might hypothesize that proximal limb elements are less responsive to locomotor selection pressures than distal limb elements, which likely encounter larger forces due to their proximity to the substrate. However, this is not the case in other mammalian orders. For example, in Primates, the pelvis exhibits low levels of integration, high levels of modularity (Grabowski, Polk & Roseman, 2011; Lewton, 2012), and differences in shape that correlate with locomotor mode (e.g., Lewton, 2015a; Lewton, 2015b; Ward, Maddux & Middleton, 2018). One way to address this question would be to examine adaptation, integration, and modularity within and among multiple hindlimb elements to discern whether there are proximal-to-distal patterns. These data will be critical to developing an explanation of the drivers of pelvic evolution within Carnivora.

Conclusions

We tested the hypothesis that carnivoran species differ in 3D shape of the pelvic bones according to locomotor function. In a taxonomically diverse sample of carnivorans, we used 3D geometric morphometrics and phylogenetic comparative methods to assess the phylogenetic, functional, and size-related effects on 3D pelvis shape. Our analyses revealed differences among taxa in pelvic shape related to size and phylogeny, but not locomotor function. These findings are similar to those of previous researchers who have found that 3D shape of the appendicular skeleton does not exhibit clear relationships with locomotor function (Martín-Serra, Figueirido & Palmqvist, 2014a; Martín-Serra, Figueirido & Palmqvist, 2014b). Our study highlights the effects of body size and allometric requirements on skeletal biology and draws parallels with previous research on pelvic bone allometry and locomotor function in the Order Primates.

Supplemental Information

Data S1 Individual landmark files for each specimen (in morphologika format) for import into R

Click here for additional data file.

Supplemental Information 1 R code

Click here for additional data file.

Table S1 List of specimens and raw, unadjusted 3D landmark data

Landmark coordinates are listed sequentially according to the order listed in Table 2: V1, V2, and V3 are the X, Y, and Z coordinates for Landmark 1; V4, V5, V6 are the X, Y, and Z coordinates for Landmark 2, and so on.

Click here for additional data file.

File S1 Carnivoran phylogeny in nexus format

Click here for additional data file.

We thank Jim Dines and Jorge Velez-Juarbe at the Department of Mammalogy at the Natural History Museum of Los Angeles for specimen access.

Additional Information and Declarations

Competing Interests

Author Contributions

Data Availability

The authors declare there are no competing interests.

Kristi L. Lewton conceived and designed the experiments, performed the experiments, analyzed the data, prepared figures and/or tables, authored or reviewed drafts of the paper, and approved the final draft.

Ryan Brankovic, William A. Byrd, Daniela Cruz, Jocelyn Morales and Serin Shin performed the experiments, authored or reviewed drafts of the paper, and approved the final draft.

The following information was supplied regarding data availability:

The raw 3D landmarks on which all analyses were performed are available in the Supplementary Files. Data S1 is for use with the provided R code (available in Code S1). The phylogeny used here is available as File S1.

3D surface models are available at Morphosource: P885: Carnivoran pelvic bones.

A list of the 55 specimens studied here (including accession numbers) is available in Table S1. All specimens are located at the Natural History Museum of Los Angeles.

https://www.morphosource.org/Detail/ProjectDetail/Show/project_id/885.

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
