# Peer review of "The effects of phylogeny, body size, and locomotor behavior on the three-dimensional shape of the pelvis in extant carnivorans"

_PeerJ, doi:10.7717/peerj.8574_

## Round 0.1 · original submission · Minor Revisions

· Academic Editor

Minor Revisions

The reviewers have provided a number of detailed comments for consideration during revision. Please address them either via inclusion in the manuscript and/or rebuttal in the response letter.

- Please also include the data files for landmarks and the phylogeny file in their original formats, so that the files can be more readily re-used by others. i.e., also include a NEXUS file for the phylogeny as well as the plain-text file used for import into R for the landmarks.
- Optionally, can the full 3D data be archived at an appropriate institutional or community repository (e.g., MorphoSource)? I realize that institutional restrictions may preclude this, but I did want to throw this out there as one option that would maximize the utility of this dataset.
- Re: the issue of specimen sex raised by one of the reviewers, I note that you used a mixed sex sample and this is stated in the supplemental information, so that aspect of the reviewer comment can be disregarded. However, I do agree with the reviewer that the impact of individual sex on the sample should probably be at least mentioned briefly.

·

Basic reporting

One issue I noticed was is in Table 1, where the list of taxa studied is accompanied by a listing of behavioral references. That list really includes few primary sources of behavioral information, and instead cites quite a few other ecomorphology studies. I think it is important to list more primary or secondary sources for those ecological attributes (i.e. behavioral studies or even mammalian species accounts). For example, Nowak (1999, or even better 7th edition from 2005) is an appropriate source for most taxa without more detailed primary info, and Sunquist and Sunquist 2002 is a very appropriate source to list for ecology of the Felidae. As a useful reference, Samuels et al. (2013) had a sample that included all species studied here, and the locomotion references listed in Appendix 1 of that study could easily be incorporated into Table 1 here.

Some of the reporting in the results would be clearer if rephrased and expanded. In particular, in lines 162, 163, 164, 173, and 174 all describe principal components as separating taxa. However, PCA does not really "separate" things, it creates multivariate axes that summarize primary sources of variation in the data set. I think description of dispersal in morphospace or variation along PC axes would be best here, which is really what the authors have done in line 167. For example, I would recommend the following for lines 162-163: "PC1 reflects variation in ischiopubic shape, and describes 31% of sample variation. PC2 relates to ilium width and orientation, and describes 23% of sample variation". For each PC axis discussed, I would recommend the authors be certain both the distribution of taxa in morphospace and morphological variation along the axis are clearly described. For example, negative PC1 scores are seen in ursids, Ailurus, and Procyon, in addition to the only skunk in the sample Mephitis. For PC3, most canids have positive scores and most felids and herpestids have negative scores, and negative scores reflect a longer ischium, while positive scores reflect a longer ilium. For PC4, while Paradoxurus is most distant from other taxa in morphospace, there are other patterns worth mentioning; specifically, most canids and felids have positive PC4 scores, while Mephitis and most herpestids have negaive scores.

Similarly, in the PGLS results (lines 182-186), I would avoid saying the analysis "demonstrated that neither locomotor mode nor size had a significant effect on pelvis shape". As written, I think that could be misinterpreted by readers, I would recommend rephrasing those sections (lines 183, 186) as "The PGLS model demonstrated that neither locomotor mode nor size had a significant effect on PC1..." and "The PGLS models demonstrated that locomotor mode did not have a significant impact on PC2 or PC3..., but size did have a significant effect on PC2 and PC3...".

Experimental design

The abstract (line 41) indicates "37 species of carnivorans" were studied, but both the methods and Table 1 list only 33 species.

In the methods (line 117) the authors indicate that data were collected on the "right os coxae of adult specimens", but later (lines 134-135) it indicates that "For specimens for which data were collected on a left os coxae...". I would recommend simply indicated up from (lines 117-118) that data were preferentially collected for right os coxae, but some specimens required collection for left os coxae and subsequent mirroring of data.

Another important point in Table 1 is that Nandinia binotata is listed among the Viverridae, but that taxon has been placed in its own family (Nandiniidae) by various recent studies. That arrangement is what was used by Nyakatura and Bininda-Emonds 2012, which the phylogeny used in this study is based upon.

The authors include 33 species (from 11 families) and analyze 6 locomotor groups. However, among the sample studied there are only two groups (terrestrial, scansorial) with a relatively diverse set of taxa, and those two groups are the least specialized. The natatorial group has one species, the cursorial group includes 3 canids (which have been hypothesized to be ancestrally cursorial in Caninae) and one felid, and the semi-fossorial group has one mephitid and one herpestid. I understand gathering the sort of data used here is challenging, but most of the phylogenetic groups used in this study are not that ecologically diverse (Table 1). As such, I would wonder if the authors would really expect to see locomotion as a primary source of variation in the data set (principal components analysis) or to have much impact of locomotion in the PGLS analyses.

Validity of the findings

No Comment

Additional comments

The manuscript was well-written and interesting, the figures were of high quality, the tables were appropriate (but see comments above), and the text generally included a thorough inclusion of relevant literature. A few other points to consider follow:

The results are compared to the findings of a number of other recent studies using 3D geometric morphometric methods (Martin-Serra et al. 2014). While this study used a more detailed documentation of pelvis morphology (as indicated in the text) and sampled some additional families, I think it is important to point out that the Martin-Serra et al. 2014 study actually had a larger sample of species. Not only were there more taxa, but the sample also had more locomotor specialists, including more cursorial canids and hyaenids, and also a more diverse sample of mustelids.

In the discussion (lines 235 - 246), the authors make comparison to some studies of primates and discuss orthograde postures. I do think it is important to note that while ursids do not typically use orthograde postures (as stated in line 245), bears seem much more capable of those postures than do most other carnivorans. While it may be speculation, pointing out that the family may have a pelvis well-suited for facultative use of orthograde postures.

The authors discuss the lack of evidence for morphological adaptations for locomotor habits in the pelvis of carnivorans, and how this may relate to constraint, I would like to see more discussion of other aspects of the postcranial skeleton as well. Among mammals there seems to be a general pattern of more pronounced modification of more distal appendicular elements. That may be a consequence of developmental, structural, or some other evolutionary constraint, but it also may be related to how organisms actually move their bodies. The distal elements of the body are the ones that interact with substrates and modification of distal elements can have profound impacts on mechanical advantage or velocity of limb movements, while not necessarily being so strongly linked to direct support of the body..

·

Basic reporting

I think it is a rather good idea to use 3D surface reconstruction to study the overall shape of the pelvis in carnivorans. I think authors made quite a good job with their paper that could meet PeerJ journal standards pending several corrections that are suggested below.

Authors use clear and unambiguous, professional English and conform to professional standards

Relevant prior literature is appropriately referenced.

The structure of the article conforms to standard sections format.

Figures are relevant to the content of the article, of sufficient resolution, and appropriately described and labeled.

Appropriate raw data have been made available in accordance with PeerJ Data Sharing policy. Some should be added such as the origin of the specimens if known (captive or wild caught).

All results are relevant to the hypothesis.

Experimental design

This paper brings relevant results in the field of morphofunctional study of the post-cranial system in carnivorans. Ecological and behavioral stressors impact on the pelvis studies are starting to bring new understandings of the role of this poorly described girdle.

Nevertheless, methods could be described with more details (see specific comments). Data are properly shared to replicate.

Validity of the findings

The manuscript brings novel results on pelvis 3D shape in relationship with phylogeny, size, and locomotor behavior. The sample size is relevant for covering a large number of taxa but, unfortunately, there are quite a few numbers of specimens for each taxon. That brings a poor statistical reliability to assess the impact of locomotor behavior on the pelvis shape, especially because intraspecific variability is not considered. Furthermore, intra-individual variability is forgotten when authors built their study on the right side of the pelvis and switch to the left side for some specimens. The impact of the specimen sex on the overall shape of the pelvis, which could be important, was not studied either on the sample. See specific comments below.

Some topics are discussed by the authors,(e.g. captive origin of the specimens) without being stated in methods and results.

However, conclusions are well driven and linked to the original research question.

Additional comments

Title: I would suggest changing “locomotion” for “locomotor behavior”.

L 33, L 35, L 37 : Try to find a synonym for “previous” to avoid the repetition (4 times).

L 88-89 This has already been explained in the first paragraph.

L 100: change “Martín-Serra and colleagues (2014a)” to “Martin-Serra et al. (2014a)”

L 104: change “Martín-Serra and colleagues (2014a)” to “ Martin-Serra et al. (2014a)”

L 111: The sample is rather interesting with a lot of taxa but shows a rather poor number of specimens per taxon (half of the species are represented by one specimen). This might be a problem to evaluate the variation of the overall pelvis shape in relationship with phylogeny, body mass and locomotor behavior since the intraspecific variation is not taken in account. Furthermore, you do not mention the sex of the specimens. There might be a risk of intraspecific variation in the pelvis shape in relationship with the sex of the individual, especially in a group of mammals. How did you deal with this question?

L 117 : you mention that data were collected on the right os coxae and then, line 135, you mention that some data were collected on the left os coxea ? How many specimens are you talking about? Which specimens are they? I found these data neither in Table 1 nor in the raw data table. As far as I know, there are quite a few intra individual variations in the articular surfaces of the pelvis and we can assume that there might be an intra-individual variation between the right and left coxal bones. Did you test this difference? This should be explained in Methods section.
Refer to:

Pallandre, J. P., Cornette, R., Placide, M. A., Pelle, E., Lavenne, F., Abad, V., ... & Bels, V. L. (2019). Iliac auricular surface morphofunctional study in Felidae. Zoology, 125714.

L 216: change “Martín-Serra and colleagues’ “ to “Martin-Serra et al. (2014a)”

L 220 : change “forelimb by (Martín-Serra et al., 2014b)” to “forelimb by Martín-Serra et al. (2014b)”

L 242 : change “(e.g., Davis, 1964; (Martín-Serra et al., 2015)” to “(Davis, 1964; Martín-Serra et al., 2015)”

L 243 : change “(Davis, 1964, p. 110)”.to “(Davis, 1964)”.
L 245-246: I would not be so directive with this conclusion. First, because ursids are known to practice bipedalism in a facultative way. Among the heaviest carnivorans species they are able to stand up and to step on hind limbs in several situations such as fighting, feeding, and threat posture…
Refer to :

Tuttle, R. H., Webb, D. M., Tuttle, N. I., & Baksh, M. (1992). Footprints and gaits of bipedal apes, bears, and barefoot people: perspectives on Pliocene tracks. Topics in primatology, 3, 221-242.

Zimmerman, A. M. (2013). Comparative Digital Examination of the Talocrural (ankle) Joint Provides Insight into Human bipedal locomotion (Doctoral dissertation, Case Western Reserve University).

Harcourt-Smith, W. E. (2015). Origin of bipedal locomotion. Handbook of paleoanthropology, 1919-1959.

Second, because ursids share, in different ways, common features with orthograde mammals (e.g. lumbar morphology).
Refer to :

Russo, G. A., & Williams, S. A. (2015). Giant pandas (Carnivora: Ailuropoda melanoleuca) and living hominoids converge on lumbar vertebral adaptations to orthograde trunk posture. Journal of human evolution, 88, 160-179.

Shapiro, L. J., & Russo, G. A. (2019). The Lumbar Spine. Spinal Evolution: Morphology, Function, and Pathology of the Spine in Hominoid Evolution, 51

L 249-250: “In addition, we used both captive and wild-reared animals in an effort to increase sample sizes“. I agree with this willingness to increase sample size but this was not mentioned in methods. This might be a bias in your study. Due to bone plasticity, because locomotor behavior and foraging strategy are dramatically reduced, captivity might impact the general pelvis shape in captive specimens. I think it would be worth it to mention how many specimens of the study are from wild or captive origin and to give the percentage of captive/wild caught in methods. You should mention this origin, when you know it, in your raw table too. You could then discuss the influence of this ratio on your results.
Refer to:

Hollister, N. (1917) Some effects of environment and habit on captive lions. Proceedings of the United States National Museum, 53, 177–193

O'REGAN, H. J., & Kitchener, A. C. (2005). The effects of captivity on the morphology of captive, domesticated and feral mammals. Mammal Review, 35(3‐4), 215-230.


Table 2
May be this is due to PDF building but landmarks (Ln) are not always in front of the definition. Can you check this please?

L2: “the opint in the center” should be “the point in the center”

C3: “to point #8” should be “to point 8”

Supplementary data:
The phylogenic tree word file is not really readable for me. Could you provide a more readable cladogram?

·

Basic reporting

Raw data: The assumption here is that the data provided are the raw and unadjusted landmark configurations. However, that is not absolutely clear and could be made more explicit. If the landmarks provided are not the raw landmarks then the authors should also include a column for centroid size.

Experimental design

This study expands on multiple prior studies. However, most specifically it expands on the work of Martín-Serra et al. (2014) who sampled 11 families and ~ 53 carnivore genera. In this study Lawton et al. sample 10 families and 33 carnivore genera and the authors indicate in lines 104-108, that their methodology expands on this prior work by first capturing anatomical regions of the os coxae not previously sampled, and via the inclusion of three carnivore families not sampled by Martín-Serra et al. (2014) (herpestids, mephitids and viverrids). The inclusion of these additional anatomical regions as well as increased sampling across the Carnivora may make sense to those who explicitly study comparative carnivore skeletal morphology, but even in those cases that would require speculation by a reader familiar with the topic as to why these differences are important. The rationale for this study would be significantly strengthened by providing explanations as to why the additional anatomical data might be advantageous (i.e. capturing relevant regions of muscle attachment or joint articulation that are critical during locomotion) and why the inclusion of these families (improved sampling at the smaller body size range?) benefits the dataset and analyses.

In the methods, the authors state that they formulated species means of Procrustes coordinates. Can the authors clarify if they created species means when: 1. Multiple specimens for that species existed, 2. If by means, they meant that they produced mean landmark configurations for each species and 3. If they generated the means from the raw landmark data and then Procrustes transformed or if they Procrustes transformed and then generated the means (and explain why).

The authors state that they used centroid size as an estimate of overall body size, which makes logical sense since the os coxae is a major weight bearing skeletal component and larger taxa should have larger ox coxae (and centroid sizes for that structure) and vice a versa for smaller taxa. However, providing some/any evidence for the correlation between body size and pelvic size in mammals seems appropriate. Either from the literature or if for at least a small subset of these museum specimens for which body size data are available and correlations between cerroid size and a body size metric could be calculated.

Validity of the findings

no comment

Additional comments

This study os well thought out and orchestrated while being well placed in the context of existing research on carnivore pelvic morphology and thus represents both meaningful replication of prior work as well as an expansion of that work. I enjoyed reading this well written article very much and think it will make a great addition to this body of work. Most of my comments are minor and strive primarily for explansion of both rationale and context by the authors with some clarification of the methodology.

Below is a final minor suggestions that I believe might make the manuscript more broadly accesible.

Starting at line 79 and elsewhere, there is an introduction of the concept of “high locomotor loads". A brief explanation of what is meant by this term would be helpful.

---

## Round 0.2 · accepted · Accept

· Academic Editor

Accept

Thank you for your close attention to the reviewers comments and editorial comments.